# Controversy in Electromagnetic Safety

**DOI:** 10.3390/ijerph192416942

**Published:** 2022-12-16

**Authors:** Chung-Kwang Chou

**Affiliations:** C.-K. Chou Consulting, 4615 Rimini Ct., Dublin, CA 94568, USA; ck.chou@ieee.org; Tel.: +1-(954)-608-7219

**Keywords:** electromagnetic fields, safety, health effects, biological effects, research, standards, regulations, risk communication

## Abstract

The dramatic increase in electromagnetic fields (EMFs) in the environment has led to public health concerns around the world. Based on over 70 years of research in this field, the World Health Organization (WHO) has concluded that scientific knowledge in this area is now more extensive than for most chemicals and that current evidence does not confirm the existence of any health consequences from exposure to low-level electromagnetic fields. However, controversy on electromagnetic safety continues. Two international groups, the International Committee on Electromagnetic Safety of the Institute of Electrical and Electronics Engineers (IEEE) and the International Commission on Non-Ionizing Radiation Protection, have been addressing this issue for decades. While the goal of both groups is to provide human exposure limits that protect against established or substantiated adverse health effects, there are groups that advocate more stringent exposure limits, based on possible biological effects. Both biological and engineering complexities make the validity of many EMF studies questionable. Controversies in research, publication, standards, regulations and risk communication concerning electromagnetic safety will be addressed in this article. The WHO is conducting systematic reviews on the RF biological effects literature. If scientists would discuss the safety issues of EMFs based on validated scientific facts and not on unreproducible possible effects and opinions, the controversy would be minimized or resolved.

## 1. Introduction

The content of this paper is condensed from a Distinguished Lecture at the Institute of Electrical and Electronics Engineers (IEEE) Broadcast Technology Society, which was held online during the pandemic, on 4 May 2021. Subsequent similar presentations were made for the Department of Energy and the National Aeronautics and Space Administration of the United States. The purpose of this paper is to convert these presentations into a printed article explaining why there is so much controversy regarding electromagnetic safety. After a summary of the history of the concerns, some key concepts are discussed. My own experience at the beginning of my research career will help to explain the origin of the controversy. In addition, examples of problems in research, publication, standards, regulations and risk communication will be given. The World Health Organization’s (WHO) effort on systematic reviews of the literature is also described. The issues are summarized, and the conclusions include hope for a better solution to minimize the controversy.

## 2. History of Issues

After World War II, in the 1950s to 1960s, the United States military initiated research programs addressing the question of whether or not radar could cause health effects to nearby personnel. In the 1960–1970s, civilians also worried about possible health effects from AM (amplitude modulation), FM (frequency modulation) and TV (television) broadcasting antennas. The introduction of microwave ovens into families in the early 1970s also caused safety concerns. In the 1980s, there were a few legal cases against radar gun companies from male policemen who had developed testicular cancer. In the 1980s there was also concern about the safety of power lines because of a report on increased leukemia in children. From the 1990s onwards, most concerns have been about wireless communications due to the prevalence of mobile phones, base stations, Wi-Fi in most households, smart meters for electricity use monitoring, radiofrequency identification (RFID) and many wearable radiofrequency (RF) devices. During the last 10 years, wireless power transmission, such as the kind used for charging cars, became popular. It can involve quite high-level RF energy levels. 

## 3. Established Scientific Understanding

Most scientists believe: There have been more than 70 years of research;Nonionizing electromagnetic radiation is dangerous only at high intensities;There is no proof of any mechanism of harm other than (1) heating from high intensity radiofrequency fields and (2) electrostimulation from low frequency fields;Non-thermal (or low-level) effects are either not repeatable or have no proven health effects;Overall, worldwide expert groups and health authorities agree that the two international exposure standards (International Commission on Non-Ionizing Radiation Protection (ICNIRP) and IEEE) are protective;Safety standards already have adequate safety margins.

## 4. Common Understanding

Unfortunately, the most convenient way for the general public to obtain knowledge is not from reading scientific journals, but from the media—either through TV, newspapers or the internet. This has resulted in a very different understanding of the subject. For reasons explained in later sections, the common understanding of the public on the biological effects of electromagnetic energy is as follows:We do not have enough understanding of the effects of electromagnetic fields (EMFs);Many reports show non-thermal effects (i.e., effects at levels below the international standards limits) (e.g., [1,2,3]);Electromagnetic radiation is dangerous;Radiation can cause cancer and many other diseases;The standards are not protective;We need precautionary measures to be safe rather than sorry.

## 5. Key Concepts

As is pointed out by the World Health Organization, although the actual public health effects ranked in order of severity are in the sequence of ultraviolet (UV) radiation, radon, X-rays and finally non-ionizing electromagnetic fields, public concerns are just the opposite, with EMFs as the highest concern. Note that UV radiation, radon and X-rays are ionizing radiation. People are afraid of “radiation” in general because they know that radiation, such as from atomic bombs and nuclear power plant accidents, can cause cancer and kill people. The general public has difficulty understanding the difference between the highly energetic ionizing radiation associated with atomic bombs and nuclear power plants and the much less energetic radiation, called non-ionizing radiation, emitted by mobile phones, base stations, Wi-Fi, smart meters, microwave ovens, radar and electrical power lines; this difficulty is a major factor contributing to the EMF controversy. The warning signs of nuclear (ionizing) radiation and RF (non-ionizing) radiation are similar as both use a yellow triangle, but one has an ionizing radiation symbol and the other has an antenna beaming waves, symbolizing non-ionizing radiation. There are even RF antenna sites mistakenly posting nuclear radiation signs to scare away people. Some merchants incorrectly use the nuclear radiation label for products related to mobile phones. It is not surprising that the general public cannot distinguish between the two very different types of radiations.

The electromagnetic spectrum covers frequencies from 0 Hz to 10^18^ Hz. The major dividing frequency is at UV-C band, when the photon energy reaches 10 eV. Above this level, the photon energy is high enough to knock electrons out of their orbits around a molecule, resulting in two electrically charged entities; this process is called ionization, hence the name ionizing radiation. EMFs at frequencies that do not have sufficient photon energy to cause ionization are known as non-ionizing radiation, which includes visible light, terahertz frequencies, radiofrequencies (including millimeter waves and microwaves), power line emissions, and static electric and magnetic fields.

Ionizing radiation has been studied since the 1890s and confirmed health effects include DNA breaks and genetic damage which can lead to cancer [4,5]. Effects due to ionizing radiation can occur from cumulative exposures. For non-ionizing (including RF) radiation, research efforts started mainly after World War II, some 70 years ago. EMFs below ultraviolet light frequencies have low photon energy, so are insufficient to cause effects such as ionizing radiation. The only confirmed adverse RF health effects relate to tissue heating due to RF energy absorption at levels that are well above the human exposure limits. Additionally, unlike ionizing radiation, there are no proven chronic or cumulative health effects of non-ionizing radiation. Most people are not aware that diathermy has been used in rehabilitation medicine for relieving joint or muscle pain in millions of patients since 1946. Diathermy utilizes RF energy to induce an increase in temperature in the tissues selected for treatment. The heating that results from RF energy absorption is responsible for the beneficial medical effects on joint and muscle pain. RF hyperthermia (RF heating) used for cancer treatment since late 1970s is another similar application for a different disease.

## 6. Steps to Address Safety Concerns

To find answers for questions requires scientific investigation or research. After conducting the research, results are often published. The results may be reviewed by an expert committee responsible for developing or revising limits in human exposure standards and, finally, regulators adopt standards into regulations for implementation. During this process, risk communication with the general public is usually through the media. In this paper, problems with each of the five steps: research, publication, standards, regulations and risk communication will be discussed.

The WHO and the International Agency for Research on Cancer (IARC) categorize 4 major types of research studies: epidemiological, human, in vivo (laboratory animal) and in vitro (cells or isolated tissues). The importance of the studies on assessing human health effects is also in the same sequence. Each of the four studies has its own advantages and disadvantages:Epidemiological studies investigate distribution of disease in human populations and factors affecting disease but can be subject to bias and confounding factors (e.g., [6]);Human studies measure the response of people to an agent, such as RF, but often have short-term exposure and subject selection (usually on healthy volunteers) (e.g., [7]);Animal studies involve the responses of large numbers of mammals and can provide faster responses compared to human and epidemiological studies. However, differences in metabolism, physiology, lifespan, etc. between laboratory animals and humans should be considered (e.g., [8]);In vitro studies have the least weight for public health evaluation but have advantages of rapid, inexpensive testing and the study of possible interaction mechanisms. However, results using in vitro systems may not be applicable to humans (e.g., [9]).

After 70 years of research, the WHO has the following statements on its website concerning EMFs biological effects or health effects [10]: “Scientific knowledge in this area is now more extensive than for most chemicals.” and “current evidence does not confirm the existence of any health consequences from exposure to low level electromagnetic fields”. Here, “low level” means levels below the current international exposure limits.

## 7. My Observation of Controversy

Figure 1 shows the source of the controversy. In the middle box, publication quality varies from A to F (Modified from [11]). The validity of the publications can be classified into 6 categories A–F. Category A publications are confirmed and established science and have the highest validity; the results have universal agreement and can be labeled as facts (as shown on the right), which can be proven and are always true, e.g., peer-reviewed published results demonstrating that high levels of RF energy cause heating which can cause adverse health effects. Professor Herman Schwan of the University of Pennsylvania, who was the pioneer of Bioelectromagnetics in the USA, once said “Good science is never outdated”. Category B is unconfirmed or to be confirmed reports. C is unconfirmed results that contradict the results of A. There are many unconfirmed D reports with clear flaws and artifacts, which are very common in this field. A few examples will be shown later. Categories E and F are unacceptable reports either in the peer-reviewed or non-peer-reviewed literature, such as newspapers or magazines. Categories B–F are papers with small to big questions marks of possible effects, as shown in the right column, and can be best classified as opinions until they can be proven to be true by independent laboratories. As shown at the top of the figure, the current controversy on biological effects of electromagnetic fields is therefore a conflict between established vs. possible effects, and thus between facts and opinions. The fundamental issue is the “quality of science”. 

I first experienced the EMF controversy as a graduate student at the University of Washington in 1971 under the guidance of Professor Arthur W. (Bill) Guy of the Department of Rehabilitation Medicine. In 1970, Mr. Jack Anderson reported in the Washington Post that the Soviets were beaming microwaves at the American Embassy in Moscow. At that time, the Soviet RF standard [12] was 1000 times lower than the US limit [13]. The US Food and Drug Administration (FDA) was interested in knowing the reason(s) for the difference in exposure limits and what the Soviets knew that we did not about microwaves. This led to research funding give to our laboratory for a project to investigate EMF biological effects, as reported by the Soviets. Dr. Guy asked me to check on the claim by the Soviet scientists that the nervous system was the most sensitive tissue to microwave “radiation”. While reading the available Soviet literature, I found that papers from the medical schools claimed all kinds of beneficial effects and therefore many medical applications, including use of high frequency RF waves in the millimeter band (now used for high band 5G wireless communication), while papers from the hygiene or environmental schools reported many harmful effects. 

My first task was to check on a 1964 Soviet study by Kamenskii [14] reporting that isolated nerves exposed in a microwave waveguide, even at relative low intensities, exhibited significant effects on the conduction velocity and excitability of the action potential transmission. I designed a temperature-controlled waveguide to show that the effects observed by Kamenskii were due to the temperature rise from absorption of RF energy in the nerve that was not detected by them because of their erroneous measurements. My research [9] showed that with proper temperature control, action potentials or muscle contractions were not affected, even with extremely high intensity radar pulses much higher than those used in the Soviet study [14].

The second part of my PhD dissertation was to check on an observation concerning the hearing of radar pulses. An American scientist reported that he had recorded neural responses, but no cochlear microphonics, in animals exposed to pulsed RF fields; these results led him to agree with the Soviet view that radar pulses can stimulate nerves, in this case the auditory nerve [15]. In my research, with a proper exposure system and shielding, I was able to record cochlear microphonics in guinea pigs and cats; these results disproved the earlier results from the American scientist that claimed pulsed RF fields stimulated the auditory nerves, because cochlea response occurs before the auditory nerve response, which leads to the auditory perception of the pulses at the cortex. For details see my tutorial and review article on this subject [16]. 

From the above two studies, I observed firsthand the EMF controversy during my graduate study in the early 1970s. Then later in my career, I started to realize more about why there is an EMF controversy. One reason was the biological diversity of my research on in vivo and in vitro studies involving various animals (frogs, mice, rats, guinea pigs, cats, rabbits, birds, pigs and monkeys) and humans, and in vitro studies on suspended cells, monolayer cells or isolated tissues [17]. I realized the complexity in biological studies that involve species, sexes, ages and extrapolation from in vitro to in vivo and to humans. In the meantime, the complexity of engineering studies is far more than the common understanding of biologists. Factors include exposure systems, far fields, near fields, dosimetry, resonance, modulation (CW (continuous wave), pulsed, AM, FM, TDMA (Time Division Multiple Access), CDMA (Code-division multiple access), LTE (long-term evolution), 5G (5th generation), experimental artifacts and temperature control (for a review, see [18]). The two complexities from biological and engineering studies make the research very complicated. Unbalanced research ability in either biological science or engineering expertise (or if both are weak) makes dealing with the complexities difficult. Like a boat with uneven rowers on each side which can only make the boat continually row in circles. 

## 8. Problems in Research

Here are a few examples to illustrate the controversy in EMF research.

Due to the large difference in limits, a United States (US)—Union of Soviet Socialist Republics (USSR) exchange program in EMF research was conducted from the mid-1970s through to the 1980s. However, the research laboratories in the two countries reported different results. For example, a Soviet group reported effects on cytochemical and immunological functions of rats at 10 μW/cm^2^ [1]. But our group in the USA found no effect other than a decrease in food consumption in rabbits exposed at 5 mW/cm^2^ (5000 μW/cm^2^, a level 500 times higher) [19,20]. 

Korbel and Thompson [2] from the University of Arkansas reported that a low-level 1 mW/cm^2^ (1000 μW/cm^2^) exposure caused behavioral changes in rats. However, Dr. Guy’s dosimetry analysis of the experiment showed the level of RF exposure could induce hot spots in the rats’ tongues while they were drinking from the metal sprout on the water bottle and their feet and tails were in contact with the metallic ground [21]. The specific absorption rate (SAR) could reach 185 W/kg in the feet and 37 W/kg in the tongue, definite hyperthermic levels that explain why an apparently low-level exposure could induce the observed behavioral effects occurred via a hyperthermic RF current flow, due to the exposure conditions. The behavioral effects reported by Korbel and Thompson were due to the high RF absorption induced in the animal’s body because of a poorly designed exposure system, and was not due to the low-level exposure as measured in air inside the exposure cage. 

Tattersall et al. [3] first reported low-level effects in rat brain slices at the Bioelectromagnetics Society meeting at Long Beach, California in 1999. I told him during the poster presentation that this effect could be due to heating from metallic electrodes. This 2001 paper has been cited as evidence for non-thermal effects on the nervous system. The effects were later shown by Tattersall to be caused by metallic electrode-induced heating [22]. When the metal electrodes were reoriented to 90 degrees perpendicular to the electric fields, the observed effect disappeared, even at a much higher exposure intensity. However, the corrected findings, entitled “Electrode-induced heating artifacts in brain slices exposed to radiofrequency fields”, had only been presented by Tattersall at a 2007 IEEE International Committee on Electromagnetic Safety (ICES) Technical Committee 95 meeting and not published in a peer-reviewed journal. Thus, the 2001 paper remains in the peer-reviewed literature as evidence for a low-level EMF effect, even with a known error.

Repacholi et al. [23] reported increased lymphomas in Eu-Pim1 transgenic mice exposed to pulsed 900 MHz electromagnetic fields. Groups of mice were exposed in cages inside a monopole exposure system. The dosimetry was complicated due to the multiple animals being in various positions. RF-induced current flow among animals in the same cages, when in contact, could occur, in similar fashion to the water drinking situation as in the University of Arkansas study. A verification study with 1600 mice in a Ferris wheel exposure system (15 units), with 40 individually positioned animals in a fix orientation in each unit, was designed for long-term exposure of lymphoma prone mice to 898.4 MHz Global System for Mobile communication (GSM) RF fields at SARs of 0.25, 1.0, 2.0 and 4.0 W/kg. This verification study showed no significant effects when compared to sham-irradiated animals [24]. Thus, a better designed exposure system with well-defined dosimetry had a different result from the original study.

Chou et al. [8] conducted a chronic exposure study with 200 rats (including 100 sham) exposed to pulsed 2450 MHz fields that was funded by the US Air Force. Among the 155 parameters studied, there was one significant difference on metastatic malignancy between exposed (18) and sham (5) rats. This statistically significant increase in primary malignancies in exposed rats vs. controls was a provocative finding, but the biological significance of this effect in the absence of truncated longevity was conjectural (there were 12 exposed and 11 sham rats alive on the final termination date). To follow up, the Air Force sponsored two more independent studies on cancer only and both showed no significant effect [25,26]. 

A recent large-scale project was the National Toxicology Program (NTP) study showing CDMA effects on glioma and both GSM and CDMA effects on heart schwannoma in male rats [27]. No effects were observed in female rats or in both sexes of mice [27,28]. One interesting result was that male rats had longer survival rates (68%, 55%, 50%) exposed to 6, 3, and 1.5 W/kg averaged over the whole body, than the sham exposed animals (28%). Survival rate increased along with the intensity of exposure. Another interesting result was that 0% glioma and 0% schwannoma in sham-exposed male rats were lower than the historical control of 2% and 1.3% of the two cancers. Note that the NTP study exposed animals chronically (i.e., for a lifetime) to up to 6 W/kg, for rats, and 10 W/kg for mice. These are very high intensity whole body average SARs compared to the human safety limit of 0.08 W/kg in both the IEEE standard (people in unrestricted environments) and ICNIRP guidelines (the general public). Comparison of the NTP whole body exposures of rodents to Federal Communications Commission (FCC) regulatory limits for local body exposure from cell phones (peak SAR of 1.6 W/kg averaged over 1-g tissue) is problematic. Based on the NTP reports, the study was to simulate whole-body exposure of humans to the RF fields from cellphone base stations (at much higher intensities for toxicology study)—not to simulate localized exposures to part of the human head, such as from hand-held cellphones. The two different parameters of whole-body-averaged SAR and 1-g averaged SAR, although with the same unit W/kg, but with different physical meanings, were not comparable. Both the ICNIRP guidelines and IEEE C95.1 standard have separate limits for the two exposure conditions, whole-body vs. local exposures. 

Currently, Japanese and Korean scientists are conducting a verification study with 140 male rats exposed at 4 W/kg (whole-body SAR) in an exposure system similar to the one used in the NTP studies. The final results are expected to be published in 2023. In 2011, Juutilainen et al. [29] presented a comprehensive review of animal studies on carcinogenicity of radiofrequency (RF) electromagnetic fields. They concluded: “Overall, the results of these studies are rather consistent and indicate no carcinogenic effects at exposure levels relevant to human exposure from mobile phones. This finding is consistent with the results of the majority of epidemiological studies on mobile phone users, and suggests that RF field exposure below the present guidelines is not likely to cause cancer”. When the NTP replication study is published, another review including the newer publications will be needed. 

As of 12 December 2022, the International Agency for Research on Cancer (IARC) has classified 1035 agents, mixtures and exposures based on the strength of scientific evidence of their potential as human cancer hazards. Known carcinogens (122) are in Classification 1, probable carcinogens (93) are in Classification 2A, and possible carcinogens (319) are in Classification 2B, while 501 are not classifiable (Classification 3). The IARC evaluation deals only with the hazard, not the risk. The IARC classified power line magnetic fields as a 2B carcinogen in 2001 and RF fields as a 2B carcinogen in 2011. In the BioEM 2022 meeting, Japanese researchers [30] reported the results of a survey among 63 scientists, 10 of them claimed that RF should be reclassified as 2A or 1, 13 said IARC classification should be downgraded to 3 and 18 said there was no need to change, while 22 others were in the “do not know or no comment” group. This is another example of the EMF controversy among scientists. 

Three weeks after the IARC declared the 2B classification of RF fields on 31 May 2011, the WHO released in late June the Fact Sheet #193 on “Electromagnetic fields and public health: mobile phones” [31] containing the following statement:

“A large number of studies have been performed over the last two decades to assess whether mobile phones pose a potential health risk. To date, no adverse health effects have been established as being caused by mobile phone use”.

This is another example which indicates the controversy between the IARC’s “possible” 2B carcinogen classification of RF fields, such as those emitted by mobile phones, and the WHO’s EMF Project’s claim of there being no “established” adverse health effects associated with mobile phone use.

Clinical results show the most relevant data in humans. Because the 2B classification of RF exposures was mainly due to the reported increased brain cancer in mobile phone users, head and neck tumors are the most relevant. The National Cancer Institute’s Surveillance, Epidemiology, and End Results (SEER) program provides information on cancer statistics, in an effort to reduce the cancer burden among the U.S. population [32]. The current data on brain and other nervous system cancer incidence rates show relatively flat, or slightly decreased rates, from the years 1992–2019, while mobile phone subscriptions increased substantially over the same period. Similar clinical data has been reported from Australia, Israel, Sweden and Taiwan. The reported increase of brain cancer in mobile phone users before 2010 was the reason for the IARC’s 2B classification. If the increased risk ratios were correct, brain cancer would have increased in recent years, but this is not the case according to the clinical data [33]. The authors of [33] concluded: “Our findings indicate that glioma incidence trends among men aged 40–59 years in the Nordic countries are not consistent with increased risks of moderate effect size (RR > 1.2–1.4) assuming latency up to 20 years. This means that increased risks reported in some case-control studies are implausible and likely attributable to biases and errors in self-reported use of mobile phone”.

The IEEE International Committee on Electromagnetic Safety website has a collection of “Statements from Governments and Expert Panels Concerning Health Effects and Safe Exposure Levels of Radiofrequency Energy” with 92 citations from 2010 to now [34]. In general, the statements can be summarized as follows: no adverse health effects have been confirmed below the current international RF safety guidelines or exposure standards (ICNIRP, IEEE).

To be valid, results of scientific studies must be repeatable and consistent. Unique results are not scientific (unlike in art). Any observed effects must have a reason (they must be repeatable before one can find out why) [35]. An old saying: it is easy for one man to throw a big rock into a well, but it will take many people, and a long time, to get it out. Doing the experiment right the first time is important, but it is difficult in this field due to the complexity of the studies. 

## 9. Problems in Publication

The following are some of the problems with scientific publications: Some cultures only publish papers reporting effects;Some journals are biased in publishing papers reporting effects;Many journals, even good ones, do not have reviewers with EMF expertise;Many published papers do not have sufficient details for evaluation or replication;Many peer-reviewed papers are not useful for the development or revision of human exposure limits, often due to inadequate attention paid to engineering or biological details, or both.

Here is an example of a controversial published study. In 2003, during a visit to the Tel Aviv University to observe an experiment that researchers claimed to show non-thermal RF effects on cultured cells, I brought 4-channel high resistance temperature probes (transparent to RF fields) to measure temperature rises in the culture media and found a 4 °C temperature gradient due to the RF exposure. Thus, I reported the effect was a thermal response and not a non-thermal effect. The engineering professor agreed that the experiments should be redone, but the paper was already submitted and later published in the Bioelectromagnetics Journal [36]. A letter to the editor was published to point out the problem [37], but the paper received the second-place award from the Bioelectromagnetics Society for the most influential journal paper in 2008. 

## 10. Problems in Standards

Human exposure standards should be developed based on the best available scientific information. Standards should be protective and practical to implement. There are three types of safety standards. Exposure standards are used for limiting human exposures. They are usually in two tiers: one for the general public and the other for occupational exposures. The IEEE C95.1-2019 standard classifies the two tiers as persons in unrestricted environments and persons permitted in restricted environments. Assessment standards are for radiating source compliance, either with measurements or computations. The third kind (interference standards) deals with medical devices. We will only discuss the exposure and assessment standards in this paper. 

Exposure standards: Firstly, who develops electromagnetic field exposure standards? Currently, there are two organizations: International Commission on Non-Ionizing Radiation Protection (ICNIRP) and the IEEE International Committee on Electromagnetic Safety (ICES) TC95 that have published standards recently. The ICNIRP has 14 experts, with no industry representatives, and works closely with the WHO. IEEE ICES TC95 has a large committee of about 130 members from 29 countries that is open to anyone with a material interest. It follows the IEEE Standards Association’s open and consensus process. 

ICNIRP has two guidelines to cover two frequency bands: “For limiting exposure to time-varying electric and magnetic fields (1 Hz–100 kHz)” [*Health Physics* 99(6): 818–836; 2010];“For limiting exposure to electromagnetic fields (100 kHz–300 GHz)” [*Health Physics* 118(5): 483–524; 2020].

Both can be obtained from https://www.icnirp.org/en/publications/index.html (accessed on 12 November 2022).

The first IEEE C95.1 standard, of only eight pages, was published in 1966 by the United States of America Standards Institute; the most recent version is IEEE C95.1-2019, of 310 pages. The risk profile of effects considered in the C95.1-2019 standard were: RF shocks and burns;Localized RF heating effects;Surface heating effects;Whole body heating effects;Microwave auditory effect;Low-level effects (previously called ‘non-thermal effects’).

The committee agreed that the first four items are adverse health effects and should be protected against. Microwave auditory effect is now a well-known phenomenon that has no adverse effect and therefore no need to be included with regards to protection. The last item on low-level effects was addressed by C95.1 on page 107 of the IEEE ICES standard: “Despite about 70 years of RF research, low-level biological effects have not been established. No theoretical mechanism has been established that supports the existence of any effect characterized by trivial heating other than microwave hearing [auditory effect]. Moreover, the relevance of reported low-level effects to health remains speculative”.

This C95.1 standard and other series of C95 standards are freely available for download from the IEEE GetProgram [38], supported by the Department of Defense of the USA.

The ICNIRP guidelines and IEEE C95.1 standard still have some differences, mainly in lower frequencies below 100 kHz. Exposure limits for higher radiofrequencies are mostly harmonized. Harmonization between the two groups is continuing. However, there are activist groups continually promoting the precautionary principle and demanding lower exposure limits to avoid all reportedly possible biological effects. The precautionary limit proposed by an activist group in their 2012 BioIntiative Report [39] was 0.3 nW/cm^2^ (0.0003 μW/cm^2^) which is unrealistically low for implementation without serious consequences in many aspects of modern life. It is noted that power density (W/m^2^ or μW/cm^2^), which can be measured in the air, has been used for exposure limits since the beginning of EMF standards in the 1960s. In 1982, the American National Standard Institute (ANSI) C95.1 standard [40] started to include specific absorption rate (SAR) in W/kg inside the body as the dosimetry for basic restriction of exposure limits. All researchers need to implement this dosimetry parameter in their RF bioeffects reporting [18]. 

Assessment standards: The assessment standards are mainly developed by both the International Electrotechnical Commission (IEC) and the IEEE ICES TC34 and published as dual logo standards. Therefore, there is no harmonization problem and not much controversy on assessment standards. The following are current standards on portable devices and base station compliance assessment. 

Portable devices (near field):**IEC/IEEE 62209-1528:2020** “Measurement procedure for the assessment of specific absorption rate of human exposure to radio frequency fields from hand-held and body-worn wireless communication devices—Human models, instrumentation and procedures (Frequency range of 4 MHz to 10 GHz)” https://standards.ieee.org/ieee/62209-1528/7325/(accessed on 12 November 2022);**IEC/IEEE 62704-3:2017** “Determining the peak spatial-average specific absorption rate (SAR) in the human body from wireless communications devices, 30 MHz to 6 GHz—Part 3: Specific requirements for using the finite difference time domain (FDTD) method for SAR calculations of mobile phones” https://standards.ieee.org/ieee/62704-3/7010/(accessed on 12 November 2022);**IEC/IEEE 63195-1, and -2: 2022** “Assessment of power density of human exposure to radio frequency fields from wireless devices in close proximity to the head and body (Frequency range of 6 GHz to 300 GHz)—Part 1: Measurement procedure, Part 2: Computational procedure” https://standards.ieee.org/ieee/63195-1/7357/and https://standards.ieee.org/ieee/63195-2/7717/(accessed on 12 November 2022).

Base station (far field in most cases):**IEC 62232:2022** "Determination of RF field strength, power density and SAR in the vicinity of base stations for the purpose of evaluating human exposure". https://webstore.iec.ch/publication/64934 (Accessed on 12 November 2022).

## 11. Problems in Regulations

When governments set regulations, there are mainly two approaches: to protect against established adverse health effects, or to protect against possible biological effects. As was pointed out by Repacholi [41], there can be orders of magnitude difference in exposure limits, depending upon whether the limits are determined using the hazard (established adverse health effect) threshold or the possible biological effect threshold. 

The former WHO EMF Project Chairman Dr. Michael Repacholi and the Russian Nonionizing Radiation Protection Committee Chair Dr. Yuri Grigoriev, and others, co-authored a paper on the “Scientific basis for the Soviet and Russian radiofrequency standards for the general public” [42] containing these statements: (1) “The general approach to public health protection and setting exposure limits by previous Soviet and current Russian committees is that people should not have to compensate for any effects produced by RF exposure, even though they are not shown to be adverse to health (pathological)”; (2) “Exposure limits are then set that do not cause any possible biological consequence among the population (regardless of age or gender) that could be detected by modern methods during the RF exposure period or long after it has finished”. This is an important difference from the approach used by the IEEE and ICNIRP. The difference in approach explains why the Russian and former Soviet exposure limits are low compared with the limits of IEEE and ICNIRP. 

From the website of the GSM Association [43], there is information of global regulatory status on portable devices and base stations. Because of the WHO influence, most countries and territories adopted the ICNIRP guidelines, but a few countries followed the regulations promulgated by the US FCC, based on the old IEEE and U.S. National Council of Radiation Protection and Measurements (NCRP) exposure limits. The countries that do not follow either ICNIRP or FCC can be divided into two groups. The first group (Belarus, Bulgaria, China, Lithuania, Poland and Russia) follows the former Soviet limits. The second group (Belgium, Chile, Greece, India, Israel, Italy, Liechtenstein and Switzerland) have implemented the precautionary principle. A recent publication shows how the past 25 years of extensive research on RFR (radiofrequency radiation) demonstrates that the assumptions underlying the FCC’s and ICNIRP’s exposure limits are invalid and continue to present a public health harm [44]. The controversy continues. 

When agendas and factors other than science become a part of the decision-making process, large disparities among regulations can be expected, as has happened around the world. Standards and regulations should be based on science with rationally defensible safety factors included to account for uncertainties and differences among populations. Non-scientifically based factors should not be included since these usually lead to arbitrary exposure limits. Overly restrictive exposure limits are often not practical and cannot be implemented. Some countries or local areas have to raise the lower limits to be able to fit the application of more advanced technology. For example, Brussels has to adjust the limit first for 4G then again for 5G communication. Others must acknowledge that the exposure limits do not apply to military operations, such as in Russia and China. 

## 12. Problems in Risk Communication

The last topic is on risk communication. Risk communication of wireless communication is difficult because the technology is complex. It is also impossible for science to “prove safety” for anything and this is a difficult concept for the public to comprehend. Precautionary recommendations can increase concerns. An example will be shown below. Therefore, the WHO recommends against arbitrary precautionary levels.

The general public mainly obtains knowledge from the media. There are differences between information from the media and science. Science emphasizes consensus and truth and follows general scientific laws, while the media deals with news, conflicts, opinions, etc. Both have the same common feature of saying “to be continued”. 

Media communication has problems because media reports on EMF safety issues are often not verified or reviewed due to urgency, unlike publications in most scientific journals, which require extensive reviews. Media stories often come from outspoken, so-called “experts”. The media loves using extraordinary and sensational stories to attract their audience. Due to today’s technologies, misinformation propagates fast and continuously. Rarely do corrections make the news, at least not in a prime spot. Unfortunately, the general public acquires knowledge from the media and not from scientific journals. A Taiwan news reporter pleaded to me, asking that scientists provide them with the correct scientific information so they do not mislead their readers. Therefore, scientists have a responsibility to ensure their findings are robust before publication, so as not to mislead the media. However, this is difficult to implement in a real world. Researchers, in general, are under pressure to produce more publications for their career advancement. 

Here is an example of the misuse of a scientific publication. Gandhi et al. [45] published a paper in 1996 showing distinct RF power deposition patterns in the heads of 5- and 10-year-old children compared to adults. A figure in the paper has been used by many activists in their websites to warn parents not to let children use mobile phones because the figure shows that exposure to the smaller heads of children resulted in deeper RF energy absorption. I raised a question about the figure to Professor Gandhi in 1999 because near field exposure should not generate such a vast difference in power deposition patterns. A 2002 publication of Gandhi and Kang [46] showed the corrected patterns with similar power absorption patterns in the adult and children heads as expected. However, no activist uses the corrected patterns, and this incorrect information continues to be available from the media, internet and other media sources. 

In 2010, there was a Eurobarometer Report [47] from the European Commission. This survey had this question: “How concerned are you about the potential health risks of electromagnetic fields?” The proportion who answered “very concerned and fairly concerned” varied by country, with participants from Denmark at 16%, while participants from Greece and Italy were both at 81%. The average was 46% in the EU. The GSM Association explained that the above average countries were either due to stricter legal safety limits or exclusion zones, or countries having strong precautionary advice by their governments. The common features of the below average countries were due to the adoption of ICNIRP limits, the implementation of a compliance program and good communication with their people. These results show that more precautionary measures can increase public concern, not decrease concern. An interesting finding!

## 13. Summary of EMF Controversy

RF bioeffect research is difficult, due to the requirement for both biological and engineering expertise.Publication quality varies widely with many papers not useful for the development or revision of human exposure limits because of poor scientific quality.What research results are to be used for exposure limits (established health effects vs. possible biological effects)?Governments’ regulations often include political considerations on precaution.Risk communication by the media focusses more on the attention-getting potential of reports, usually involving only one or a few individuals, rather than scientific consensus.The general public is confused by conflicting scientific reports and unharmonized standards and regulations reported in the media.

## 14. WHO Systematic Review of Scientific Results

As indicated in Figure 1, the center of the controversy is the quality of science, i.e., the validity of the publications. Foster et al. [48], in the section of a book chapter on “Approaches to Critical Reviews of the Literature”, described ways to evaluate published results. When conducting critical reviews (different from “bean-counting”, that is, merely counting reported positive or negative effects), it is necessary to assess the quality of individual studies, searching for consistencies in findings across different studies, and the relevance of biological effects to human health. The earlier approaches of subject narrative reviews and weight-of-evidence reviews have evolved to meta-analysis and the latest systematic reviews [49,50]. A systematic review answers a defined research question by collecting and summarizing all empirical evidence that fits pre-specified eligibility criteria. Meta-analysis is the use of statistical methods to summarize the results of independent studies. These methods will reduce bias and improve the objective judgement of the published results. Evidence-based medicine has led to a widespread use of systematic review and meta-analyses. Currently, they are the commonly used methods, especially systematic review, in the WHO‘s health-related projects which assess the effects of various environmental factors. 

From October 2019 to December 2021, the WHO International EMF Project issued a call for experts (https://www.who.int/news-room/articles-detail/call-for-experts-who-task-group-on-radiofrequency-fields-and-health-risks (accessed on 12 November 2022)) to form a WHO Task Group to conduct “Appraisal of the evidence for health risks associated with exposure to RF fields” and to produce the following documents, as presented recently by Dr. Emilie van Deventer of the WHO, [51]:A Technical Report (scoping review of the scientific literature of studied health outcomes);A series of Systematic Reviews on priority health outcomes to be published in a special issue of Environmental International;An Environmental Health Criteria (EHC) Monograph that will elaborate on the health outcomes highlighted in the review process, using procedures for guideline development as recently required by WHO;A RF Research Agenda;Journal publications.

A survey to prioritize health outcomes when assessing the effects of exposure to radiofrequency electromagnetic fields has been completed as reported in [52]. The survey resulted in 10 topics which are: (1) cancer (human observational studies), (2) cancer (animal studies), (3) adverse reproductive outcomes (human observational studies), (4) adverse reproductive outcomes (animal and in vitro studies), (5) cognitive impairment (human observational studies), (6) cognitive impairment (human experimental studies), (7) symptoms (human observational studies), (8) symptoms (human experimental studies), (9) the effect of exposure to RF on biomarkers of oxidative stress and (10) the effect of exposure to heat from any source on pain, burns, cataract and heat-related illnesses. The systematic review protocols of the first 9 topics have been published [53]. The review results of these topics will be published in Environmental International. 

After the results are all in, a final RF EHC will be published as a monograph by the WHO. This will be the first EMF-related EHC based on systematic reviews. Future assessments of the biological effects of non-ionizing fields and revisions of human exposure standards are expected to be substantially dependent upon the results of these systematic reviews. Furthermore, the conclusions in the systematic reviews should greatly reduce the controversy in electromagnetic safety.

## 15. Conclusions

During the last 50 years, I have learned that electromagnetic (non-ionizing) exposure is very different from nuclear (ionizing) radiation. Seventy years of research shows excessive thermal effects are an established adverse health effect of RF energy (above 100 kHz), electrostimulation is the established adverse health effect of exposures below 100 kHz. International exposure (with large safety margins) and assessment standards are available to provide protection to both the general public and workers. A large number of expert scientific reviews have concluded that no adverse health effects have been confirmed below the current international RF exposure limits (ICNIRP, IEEE). Ordinary exposures are very low. Unnecessary worry can cause nocebo effects (if one believes there may be side effects, one may often experience such effects).

My hope for the EMF controversy is for there to be more facts and less opinions. The publication of the WHO’s Environmental Health Criteria Monograph based on systematic reviews will greatly improve the validity of future evaluations of the health effects of exposure to non-ionizing energy. If scientists would discuss EMF safety issues based on validated scientific facts and not on unreproducible possible effects and opinions, the controversy would be minimized or resolved. Confucius said, “knowing what you know and what you don’t know, that is knowledge”. Let us make safety standards and regulations on electromagnetic exposures based on what we know, and not on what we do not know for sure. 

## Figures and Tables

**Figure 1 ijerph-19-16942-f001:**
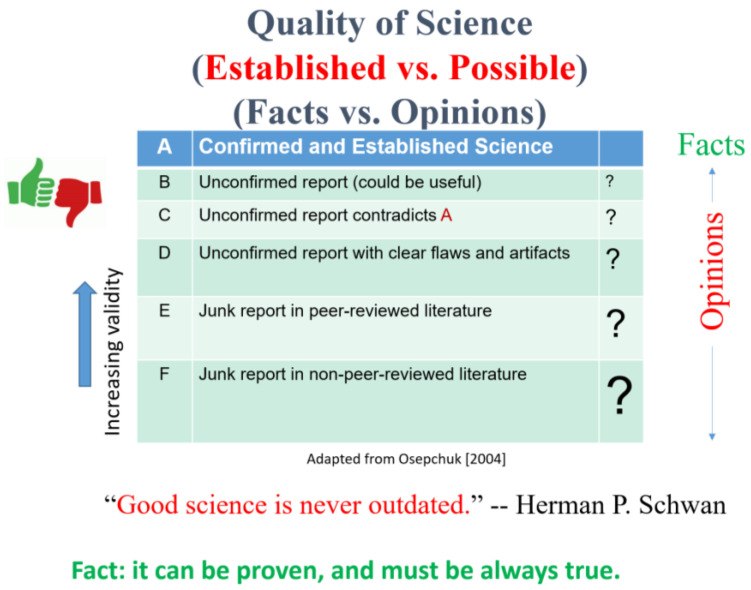
Center of the controversy: conflict between facts and opinions.

## Data Availability

Not applicable.

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
