# Peer review of "Controversy in Electromagnetic Safety"

_ijerph, 2022, doi:10.3390/ijerph192416942_

Round 1

Reviewer 1 Report

The paper considers the very important issue of public health concerns in the world according to increasing EMF exposure in everyday life.
Despite a large number of published studies and systematic reviews conducted by worldwide expert groups and health authorities, it is difficult to come to valid and unambiguous conclusions and controversy on electromagnetic safety continues.
The work is a very interesting,  popular-science work.
The Author clearly and concisely presents the current state of knowledge and a picture of the public's awareness of the impact and safety of EMF. The Author tries to define the causes of misinterpretation of published research.
The Author did not present anything new but concisely summarized the important and current worldwide problem. In addition, the Author describes his experience in his scientific work creating a historical background that adds to the appeal of the text. Articles like this are important and should be published from time to time. They should be published, not only in scientific journals but also in more popular ones, which people from the general population reach for.

Author Response

The comments of this reviewer are very much appreciated. Thank you very much for agreeing with me on the content. Hope this article will help the readers to understand why there is so much controversy. 

Reviewer 2 Report

I found the paper, which is actually a letter to the editor with perspectives on the future of RF non-ionizing radiation safety, very interesting and should be accepted.

This paper should be read by all researchers on electromagnetic fields (EMF).

Line 126 to 136, for each of the types of studies it is necessary to include cites as examples.

Figure 1 seems very useful and provides useful information, but I cannot understand it completely, I suggest comment this diagram.

Line 289, include the citation from the study of Japanese researchers.

Lines 353 to 357, include two citations, one for each of the standards mentioned (exposure and assessment).

Line 383, there is a small error, instead of hearing it should be heating

Line 398, include a citation regarding the BioIntative report.

Lines 405, 409, 413, and 418, include the corresponding citations.

Line 532, include a citation. Also include a citation of the two aforementioned documents, lines 534 and 538.

The author uses in his manuscript units of mW/cm2, µW/cm2, and nW/cm2. A paper has recently appeared that recommends using µW/m2, the author is suggested to cite it, if he considers it appropriate, and to use that unit in his article. The paper mentioned is:

Ramirez-Vazquez, R., Escobar, I., Franco, T., Arribas, E., 2022. Physical units to report intensity of electromagnetic wave. Environment. Res. 204, 112341. https://doi.org/10.1016/j.envres.2021.112341

Author Response

Comments and Suggestions for Authors

I found the paper, which is actually a letter to the editor with perspectives on the future of RF non-ionizing radiation safety, very interesting and should be accepted.

Response: Thank you. According to the classification of the journal, this article is submitted as a Perspective.  

This paper should be read by all researchers on electromagnetic fields (EMF).

Response: Thank you very much for this comment.

Line 126 to 136, for each of the types of studies it is necessary to include cites as examples.

Response: Example references on four types of studies are added.

Figure 1 seems very useful and provides useful information, but I cannot understand it completely, I suggest comment this diagram.

Response: Sorry. More text is added to explain more of Figure 1. Hope it is now more understandable.

Line 289, include the citation from the study of Japanese researchers.

Response: Added reference to the Japanese study presented at the BioEM2022 meeting.

Lines 353 to 357, include two citations, one for each of the standards mentioned (exposure and assessment).

Response: Exposure and assessment standards are listed below in this section.

Line 383, there is a small error, instead of hearing it should be heating

Response: Microwave hearing is correct, not microwave heating. This is about the hearing of microwave pulses as mentioned in Section 7. To avoid possible confusion, I changed microwave hearing effect to microwave auditory effect.

Line 398, include a citation regarding the BioIntative report.

Response: Reference of BioInitiative report is added.

Lines 405, 409, 413, and 418, include the corresponding citations.

Response: Links are included for the four standards.

Line 532, include a citation. Also include a citation of the two aforementioned documents, lines 534 and 538.

Response: One link and one reference are added.  I also updated the information based on a more recent presentation by Dr. Emilie van Deventer of the WHO.

The author uses in his manuscript units of mW/cm2, µW/cm2, and nW/cm2. A paper has recently appeared that recommends using µW/m2, the author is suggested to cite it, if he considers it appropriate, and to use that unit in his article. The paper mentioned is:

Ramirez-Vazquez, R., Escobar, I., Franco, T., Arribas, E., 2022. Physical units to report intensity of electromagnetic wave. Environment. Res. 204, 112341. https://doi.org/10.1016/j.envres.2021.112341

Response: I do not see this is a big problem. But I add µW/cm2 to each site mentioned power density unit to address this comment.

Reviewer 3 Report

The article by C-K Chou entitled “ Controversy in Electromagnetic Safety” attempts to present the main sources of disagreement in the assessment of the biological effects of  radiation of the radiofrequency range. It is a very interesting paper, well written and easy to follow by non-experts in the field. My few minor comments listed below are aiming to slightly increase the clarity of the manuscript.

Page 2, sec. 3: Pls explain what you mean exactly by “the two international standards are protective”. Also maybe briefly mention these two standards at this point.

Page 2, Sec.4: I think a few refs. at bullet #2 would be useful, since you mention reports.

Page 2, Sec. 5: Pls remove “The” before “general public”.

Page 3, line 100: Please provide a couple of  refs, i.e. some ICRP or BEIR report.

Page 4, l. 165: Pls explain which this standard was.

Page 5, l. 177-178: Pls explain what you mean by “severe effects on the action potential transmission”.

Page 5, l. 190: Pls add “the” before  ”then to the cortex”.

Page 5, l. 198: Pls explain what you mean by “strains”.

Page 6, 3rd paragraph: Pls explain the abbreviations GSM, CDMA, FCC. Pls also check the rest of the document for unexplained abbreviations.

Page 6, l. 268-271: Pls rephrase this final sentence.

Page 6, l. 259: Pls remove “exposed” before “male rats” since it is repeated in the sentence.

Page 7, l. 289-290: The sentence is not very clear. Maybe you could change to “ … among 63 scientists. 10 of them viewed…”

Page 9, l. 397: Pls explain what this report is or provide some reference (if any). Also, since this manuscript seems to be addressed also to the non-familiar with this particular field readers, I believe it would be useful to explain the main dosimetric parameters of non-ionising radiation (SAR, power density, …)

Page 10, l. 468: “EME issues”?

Page 10, l. 484: “that more of the head”. You mean larger part of the head?

Page 11, 494-501: I think it would be useful to elaborate more on the comments of this paragraph, like, how established are the precautious limits adopted? Are there recommendations (i.e.  from EU in Europe) that countries do not follow?

Author Response

Comments and Suggestions for Authors

The article by C-K Chou entitled “Controversy in Electromagnetic Safety” attempts to present the main sources of disagreement in the assessment of the biological effects of radiation of the radiofrequency range. It is a very interesting paper, well written and easy to follow by non-experts in the field. My few minor comments listed below are aiming to slightly increase the clarity of the manuscript.

Response: Thank you very much for your comments for me to improve the accuracy and readability.

Page 2, sec. 3: Pls explain what you mean exactly by “the two international standards are protective”. Also maybe briefly mention these two standards at this point.

Response: Added “international exposure standards (ICNIRP and IEEE) are protective” The 92 expert reviews in the ICES website show that most expert groups view that the two international standards are protective.

Page 2, Sec.4: I think a few refs. at bullet #2 would be useful, since you mention reports.

Response:  Section 3 and 4 are just summary of the important differences. But I added three existing references of the papers that are discussed later.

Page 2, Sec. 5: Pls remove “The” before “general public”.

Response: Deleted several “the” before “general public”, when appropriate.

Page 3, line 100: Please provide a couple of refs, i.e. some ICRP or BEIR report.

Response: Two references on ionizing radiation effects are added.

Page 4, l. 165: Pls explain which this standard was.

Response: Soviet and US limit references are added.

Page 5, l. 177-178: Pls explain what you mean by “severe effects on the action potential transmission”.

Response: Modified text to clarify.

Page 5, l. 190: Pls add “the” before ”then to the cortex”.

Response: I do not understand this comment. But I made some edits of the sentence to make it more clear.

Page 5, l. 198: Pls explain what you mean by “strains”.

Response: The main difference between strain and species is that strain is a genetic variant; it is a subtype or culture of a biological species. Since it gets too detail, I deleted the term “strains”.

Page 6, 3rd paragraph: Pls explain the abbreviations GSM, CDMA, FCC. Pls also check the rest of the document for unexplained abbreviations.

Response: Added terms for all abbreviations appear first in the article.

Page 6, l. 268-271: Pls rephrase this final sentence.

Response: Sentence rephrased to clarify that comparing the FCC cellphone exposure limit to the NTP exposure levels is inappropriate, because the study was for whole body exposure.

Page 6, l. 259: Pls remove “exposed” before “male rats” since it is repeated in the sentence.

Response: Yes, removed this unnecessary word.

Page 7, l. 289-290: The sentence is not very clear. Maybe you could change to “ … among 63 scientists. 10 of them viewed…”

Response: Revised as suggested.

Page 9, l. 397: Pls explain what this report is or provide some reference (if any). Also, since this manuscript seems to be addressed also to the non-familiar with this particular field readers, I believe it would be useful to explain the main dosimetric parameters of non-ionising radiation (SAR, power density, …)

Response: Added BioInitiative Report reference. Added some sentences on the difference between SAR and power density.

Page 10, l. 468: “EME issues”?

Response: EME is an abbreviation for electromagnetic energy. I changed it to EMF safety issues to be consistent within this article.

Page 10, l. 484: “that more of the head”. You mean larger part of the head?

Response: Clarify the sentence to mean deeper penetration into the head.

Page 11, 494-501: I think it would be useful to elaborate more on the comments of this paragraph, like, how established are the precautious limits adopted? Are there recommendations (i.e.  from EU in Europe) that countries do not follow?

Response: I added some sentences on the precautionary low limits at the end of Section 11, where talked about exposure limits in regulations. Several EU countries do not follow the EU recommendations as listed in Section 11 on regulations.